



# Possible evidence of increased global cloudiness due to aerosol-cloud interactions

Alyson Douglas[1] and Tristan L'Ecuyer[2, 3]

[1]Atmospheric, Oceanic and Planetary Physics Department, Department of Physics, University of Oxford, Sherrington Rd, Oxford OX1 3PU, United Kingdom
[2]Cooperative Institute for Meteorological Satellite Studies, University of Wisconsin-Madison, 1225 W Dayton St, Madison, WI, USA
[3]Department of Atmospheric and Oceanic Sciences, University of Wisconsin-Madison, 1225 W Dayton St, Madison, WI, USA

**Correspondence:** Alyson Douglas (alyson.douglas@physics.ox.ac.uk)

**Abstract.** Aerosol-cloud interactions remain a large source of uncertainty in global climate models due to uncertainty in how pre-industrial clouds, aerosols, and the environment behaved. We employ three machine learning models, a random forest, a stochastic gradient boosting, and an extreme gradient boosting regressor to derive a pre-industrial proxy for warm cloudiness predicted using only their environmental controls. We train our models on boundary layer stability, relative humidity of the free

atmosphere, upper level vertical motion, and sea surface temperature to predict a simulated, pristine cloud fraction as a one-for-one proxy for a pre-industrial warm cloud fraction. Using a multivariate linear regression as a proxy for sensitivity studies, we show that the non-linear signatures derived using the simple machine learning models are pivotal in deriving an accurate estimate. We find that aerosols may have increased global cloudiness by 1.27% since pre-industrial times, leading to -.42 (.39 - .46 at 95% confidence intervals) of cooling. Our methodology reduces the covariability between aerosol, the environment, and

cloud adjustments by aiming only to estimate an initial, unperturbed state of the cloud based on the environment alone.

## 1   Introduction

Aerosol-cloud interactions (ACI) are a leading source of uncertainty in future climate (Boucher et al., 2013). Due to computational limits, the effects of aerosol on cloud properties cannot be explicitly resolved by current global climate models leading to our dependence on cloud parameterizations. These parameterizations do a poor job at representing ACI and the myriad of

effects they have on the Earth system through cloud-climate interactions (Stevens and Bony, 2013). Aerosol enters a cloud and acts as cloud condensation nuclei (CCN). As the droplet number (Nd) increases for a given liquid water path (LWP), the mean droplet size decreases, increasing the cloud's albedo and therefore cooling the atmosphere by reflecting more solar irradiance in what is termed the radiative forcing due to ACI (RFaci) (Twomey, 1977). The decrease in mean droplet size also reduces the chance of collision-coalescence, suppressing precipitation and theoretically increasing the cloud's lifetime. Under certain

meteorological conditions, a delay in onset of precipitation and prolonged lifetime leads to larger liquid water paths and more expansive clouds, additionally cooling the atmosphere through prolonged cloud existence through what are termed collectively



cloud adjustments (Albrecht, 1989). While global climate models fail to accurately represent these interactions within parameterizations, using observations to quantify the behavior of clouds has also proven challenging owing the need to account for the compounding complexity of aerosol-cloud-environment interactions (Andersen and Cermak, 2015).

While there is evidence of aerosols prolonging cloud lifetime and increasing the overall cloud fraction, LWP responses remain unclear. Estimates of cooling due to changes in LWP may range from a weak warming to strong cooling, depending on the source of emissions (shiptracks, biomass burning, etc.) and the methodology used to quantify it (Bellouin et al., 2019; Rosenfeld et al., 2019). Similarly, although increased cloud extent should in theory be an obvious cloud adjustment to observe, as cloud fraction is directly observed by satellites unlike other variables that must be derived like Nd or LWP, due to confound-

ing relationships between aerosol proxies and the environment, isolating the change in cloudiness due to only anthropogenic aerosol loading remains challenging. Particularly important is understanding how ACI leads to cooling in warm clouds, the most commonly occurring cloud type on Earth, that have been shown to be extremely important to the radiative balance (Langton et al., 2021; Christensen et al., 2016; Hahn et al., 2001). In warm clouds, it has been shown using a Lagrangian framework that it is the combination of aerosols and the environment that lead to sustained cloudiness (Christensen et al., 2020). How-

ever, even with evidence of changes to cloudiness due to aerosol, it remains unclear how aerosol modifies the LWP, especially as environmental modulation of ACI may buffer the observed reaction (Stevens and Feingold, 2009). For these reasons, any methodology that evaluates cloud adjustments due to ACI must recognize that the environment is one of the primary drivers of cloud properties and quantifying the cloud response to aerosol must account for the covariability with the environment.

     Machine learning (ML) offers potential methods to deal with not just the covariability of multiple cloud adjustments with

the environment, but also the non-linear relationships between cloud adjustments and aerosols (Alpaydin, 2020). Even simple methods, like a random forest regression or stochastic gradient boosting, can be used to derive complex, non-linear relationships. In this study, we test the ability of simple machine learning methods to realistically predict cloud fraction in pristine aerosol conditions using only the environment as features for the prediction. We hypothesize a "good" ML model will be:

– Physically realistic: Any machine learning algorithm applied should either be easily constrained to be physically realistic

or able to learn the physical limitations of the real atmosphere.

– Observational scale independent: The averaging scale of observations used will not lead to a large bias in the predicted change in a cloud property (McComiskey et al., 2012).

Further, the relationships learned by the algorithm may expand our knowledge of the current processes and help inform decisions on how to "tune" cloud parameterizations within global climate models. Interpretations of the model output could offer

novel insights into how various components of the atmosphere link with aerosol to control cloud properties (Zantedeschi et al., 2019).

     Recent work has exploited the unpolluted aerosol conditions of the southern hemisphere to constraint on global climate models (GCM) estimates of RFaci (McCoy et al., 2020). We aim to similarly use present day pristine conditions to determine the state of warm cloud amount before the industrial revolution. Unlike (Chen et al., 2014), (Feingold et al., 2016), and others

who aim to quantify a sensitivity or susceptibility of a cloud to aerosol, or the linear regression of a cloud property against





aerosol concentration; we aim to estimate the unperturbed warm cloud fraction to use as a scene by scene, counterfactual, pre-industrial comparison. In certain conditions, aerosol in the marine environment can imitate pre-industrial conditions (Hamilton et al., 2014). By aiming to predict cloud fraction under pristine, natural-aerosol conditions as a proxy for pre-industrial cloudiness, rather than estimate the function that describes the relationship between aerosol and cloud fraction, the complexity of the

problem is reduced. This allows us to estimate the change in cloudiness since pre-industrial times using the predicted pristine cloud fraction, circumventing the need for a precise function that describes the non-linear relationship between aerosol, cloudiness, and the environment in which the interactions occur. Our method aims only to capture the increase in warm cloudiness due to aerosol, however other changes since pre-industrial times such as increases in $CO_2$ and induced cloud feedbacks, such as reductions in warm clouds due to higher surface temperatures, may have furthermore changed warm cloudiness (Clement

et al., 2009).

Our hypothesis and assumptions herein are simple, though important: that aerosols have increased warm cloudiness since pre-industrial times and that pre-industrial cloudiness can be approximated using present day observations of clouds occurring in pristine aerosol conditions. Although ML is a more advanced statistical method with obscure pathways to prediction than a linear regression, by utilizing simple ML models to predict and test our simple theory, uncertainty lies in quantifying how

well the models predict pristine cloudiness. The uncertainty is not predicated on the accuracy of current satellite observations of aerosol size or concentration. Our method is less prone to methodological error compared to more complicated techniques, such as those that employ kernels or regress the sensitivity of cloud properties to both the aerosol and environment at once. In a field drowning in uncertainty, clever execution of ML techniques may offer a lifebuoy to aerosol-cloud interactions (Emanuel, 2020).

## 75  2  Methods

### 3  Data

Observations of clouds, aerosols, and sea surface temperature (SST) from 2007 to 2010 are provided by instruments aboard the A-Train satellite constellation from 60°S to 60°N. Cloud fraction is estimated using observations of cloud properties from the cloud profiling radar (CPR) aboard CloudSat and Cloud-Aerosol Lidar with Orthogonal Polarization (CALIOP) aboard Cloud-

Aerosol Lidar and Infrared Pathfinder Satellite Observations (CALIPSO). The aerosol index, a proxy for aerosol concentration, is from the Moderate Resolution Imaging Spectroradiometer (MODIS). Sea surface temperature is from Advanced Microwave Scanning Radiometer for EOS (AMSR-E) instrument. AMSR-E and MODIS are both aboard the Aqua satellite, which led the A-Train constellation that included CloudSat and CALIPSO from 2006-2017 (L'Ecuyer and Jiang, 2010).

An along track warm cloud fraction is derived using CloudSat 2B-CLDCLASS-LIDAR cloud top heights along with a

freezing level defined using the ECMWF-aux product. Observations are limited to marine warm clouds defined as having a cloud top height below the freezing level and averaged over 12 km or 96 km along track segments (with CloudSat having a native resolution of ∼1 km x 1km). Observations are collocated by using the largest satellite footprint as the along-track size (12 km) with fields-of-view of the instruments overlapping along the A-Train track. By using 2B-CLDCLASS-LIDAR,





extremely thin and low lying warm clouds are captured by the CALIOP lidar (Sassen et al., 2008). The cloud fraction is found

at 12 km and 96 km scales along the satellite track in order to test how the pristine cloudiness estimates depend on the averaging scale of observations. The outgoing shortwave radiation at the top-of-atmosphere (OSR) used to calculate the cooling effect is from the 2B-FLXHR-LIDAR product and is normalized using the mean solar insolation to account for seasonality (Henderson et al., 2013).

The MODIS aerosol index (AI), defined as the product of the Angstrom exponent and aerosol optical depth at 550nm, is

interpolated between cloud-free pixels in order to approximate the amount of aerosol available to act as CCN near the warm clouds (Ginoux et al., 2012). While near cloud AI can be affected by hygroscopic growth of aerosol particles, this should not greatly affect the results of this study (Christensen et al., 2017). Our AI estimates are interpolated between 2 km from the edges of the clouds in order to deal with a majority of hygroscopic growth. We use the pre-industrial aerosol index from the Spectral Radiation-Transport Model for Aerosol Species (SPRINTARS) model's historical run to constrain a training set of observations

that approximately mimics the pre-industrial distribution (Takemura et al., 2005). Our pre-industrial AI scenario is merely a rough upper limit during training and does not drive any assumptions of pre-industrial CCN in the subsequent analysis. Our models are trained on this "clean" subset of observational data as a basis for predicting a pristine cloud fraction using only the environmental information as a counterfactual for the actual, observed cloud fraction. We choose a selection of scenes with a distribution of AI similar to the pre-industrial values for that region from SPRINTARS since there is evidence that even remote

regions, like those in the southern ocean, contain large concentrations of natural aerosol (Quinn and Bates, 2011).

Precise measurements of pre-industrial or present day AI are therefore not as important as regression based studies; we are not quantifying any relationships in terms of AI, merely using it as a constraint on our training sample. Removing dependencies on aerosol, both observed present day and modeled pre-industrial, reduces uncertainty in our forcing estimates compared to other studies which rely on the change in aerosol to quantify the ERFaci or a sensitivity. Pre-industrial aerosol is largely

uncertain, therefore it is more advantageous to employ it in limited use as a "soft" constraint on the training subset (Carslaw et al., 2013).

MERRA-2 temperature, wind, and humidity profiles are used to derive the stability, relative humidity of the free atmosphere, and winds at 500 hPa on which each model is trained to predict a pristine cloud fraction (Gelaro et al., 2017). The stability of the atmosphere is represented by the estimated inversion strength, which has been shown to be highly correlated with cloudiness as

a capping inversion usually leads to extended cloud lifetimes (Wood and Bretherton, 2006; Wood, 2012). The relative humidity of the free atmosphere is indicated by the humidity at 700 hPa, chosen to reflect the effects of entrainment of free atmospheric air through the cloud top (Douglas and L'Ecuyer, 2019). It is possible that 700 hPa may be above the entraining level for some shallower low clouds, but should generalize the impacts of large scale subsidence on the boundary layer. Vertical motion at 500 hPa from MERRA-2 reflects how large scale motions influence the cloud layer in pristine aerosol environments (Randall,

1984). MERRA-2 has been shown to have biases in its representation of the hydrological cycle, however as we are using it to approximate the boundary layer atmosphere, and not precipitation or water vapor fluxes, we do not believe MERRA-2 biases will greatly effect our results (Bosilovich et al., 2017). It is possible some error in the reanalysis may lead to a bias of the results although it should be emphasized that the reanalysis must only be accurate enough to capture the thermodynamic and





dynamic environments. Collectively these four parameters (SST, stability, $RH_{700}$, and $\omega_{500}$) provide a bulk characterization

of the local environment, essential for separating environmental drivers of cloud cover from aerosol impacts (Douglas and L'Ecuyer, 2020).

## 3.1 Methodology

Four models are used to predict a pristine aerosol condition warm cloud fraction adapted from sci-kit learn and the Python XGBoost module (Pedregosa et al., 2011; Chen et al., 2015). A bootstrapped random forest regressor (RF), a stochastic gra-

dient boosted regressor (SGB), and an extreme gradient boosted regressor (XG) are simple machine learning models that are compared to a multivariate linear regression (MVLR). Observations gridded at 15° x 15° are used to train/test/validate the regionally specific models. Only cloudy pixels ($CF_N > 0$ where N is 96 km or 12 km) are used to train the model. The models are evaluated by their mean squared error (MSE) and their coefficient of determination values. The uncertainty of the RF, SGB, XG are evaluated by comparing how the predictions vary when re-trained different subset of observations.

The non-linear relationship between aerosol and cloud amount is somewhat circumvented since the models are only tasked to predict pristine, "initial" cloud state, not to quantify the change or rate of change. Cloud adjustments remain so uncertain that it is unknown how ACI affect LWP. Therefore, by avoiding trying to quantify how aerosol affects the LWP, and by focusing on the most easily observed feature of a cloud, its extent, we reduce the uncertainty of our results. Our given forcing estimates therefore assume a constant cloud albedo and do not account for the Twomey effect or other cloud adjustments. We further

reduce uncertainty by only focusing on warm, marine clouds from 60°S to 60°N; we limit our observations to only warm phase clouds to reduce the uncertainty of aerosol-cloud interactions in supercooled liquid, mixed phase, or ice phase clouds.

The vertical motion at 500 hPa, the estimated inversion strength, and the relative humidity at 700 hPa, all from MERRA-2, are used along with the sea surface temperature from AMSR-E as the environmental features to train and predict the pristine cloud fraction (Partain, 2004). In order to delineate the training and testing datasets, observations are considered pristine if the

aerosol index from MODIS is at or below the a threshold determined by the distribution of pre-industrial aerosol index from SPRINTARS for that region. These pristine observations are then split into by 20% test size and 80% training size for each model.

All four methods examined are given the same features and warm cloud fractions to be used in the respective training and testing phases. After testing the ability of each method to predict a known pristine cloud fraction, each method is tasked

with estimating what the pristine cloud fraction would have been in clouds that lived in present-day, "polluted" environments. The difference between the two cloud fractions is the estimated change in warm cloud fraction due purely to the aerosol concentration increasing.

$$\Delta CF = CF_{actual} - CF_{predicted\ pristine} \tag{1}$$

where $CF_{predicted\ pristine}$ is the cloud fraction estimated using the RF, SGB, XG, or a MVLR.





The shortwave forcing due to a changes in cloud fraction owing to anthropogenic ACI is found by taking the product of $\Delta \mathrm{CF}_{warm}$ and the mean, normalized shortwave cloud radiative effect at the top-of-atmosphere (SWCRE) for the region.

where W is the weight is determined by the frequency of occurrence of warm clouds in that region between 2007 to 2010. The result is an estimate of the radiative forcing due to cloud adjustments that avoids estimating cloud susceptibility and

pre-industrial aerosol distributions that can be very uncertain (Carslaw et al., 2017).

The predicted pristine cloud fraction estimate depends only on the learned relationships between the environmental features and the cloud fraction. This reduces the uncertainty due to error in aerosol index retrievals near clouds. With the approach adopted here, there is some uncertainty due to aerosol-environment interactions, whereby the environment may be affected by aerosol direct effects (Matus et al., 2019). Our approach may include some semi-direct effects, such as when aerosol within

the cloud warms and desiccates the cloud layer, as we take the difference in polluted and simulated, pristine cloud fractions without controlling for where the aerosol is located or how the environment may promote or impede these semi-direct effects (Herbert et al., 2020). We do not include polar regions in our models in order to reduce uncertainty that may occur due to supercooled liquid and mixed phase clouds that are common in these regions. It is possible neglecting these clouds induces some uncertainty in our estimates of global forcing, however we believe the magnitude of changes in these regions is much

smaller than those found in warm topped, liquid phase clouds we evaluate within. The uncertainty due to direct effects should be minimal as our models are trained on pristine condition aerosols not likely to substantially heat the environment. The learned cloud fraction should approximately, and we say approximately as the Earth is slightly warmer than in the pre-industrial era, represent how clouds behaved in pre-industrial times when there were no emissions due to human activity and the environment, to a first order, controlled the cloud properties.

**3.2   Machine Learning Models**

A random forest regressor (RF), stochastic gradient boosted regressor (SGB), and an extreme gradient boosted regressor (XG) are chosen as the simple machine learning models to be compared against results from a multivariate linear regression can be compared against. Though a multivariate linear regression (MVLR) could be considered a machine learning model, it is used as a comparison to highlight how more advanced types of machine learning capture non-linear behavior unable to be represented

by an MVLR. Further, other studies have used an MVLR (such as (Fuchs et al., 2018)) with ERFaci estimates at the high end of the IPCC observational range, which we suggest are only capturing a mean signal compared to the true, non-linear signal our simple ML models present. All three simple models use decision trees as their basis (Figure 1).

A RF regressor uses a collection of decision trees to "vote" on the predicted outcome, such as the cloud fraction in our case (Liaw et al., 2002). The random forest regressor used within is trained using the mean squared error as the loss function. This

differs a RF from a SGB or XG regressor, which are fit based on the gradient of the loss function towards a minima (Friedman, 2002). An SGB regressor uses a random sample from the dataset during the fitting of the decision trees and continues to fit the residuals, while the XG regressor uses an optimized loss function to more quickly and accurately fit the decision trees





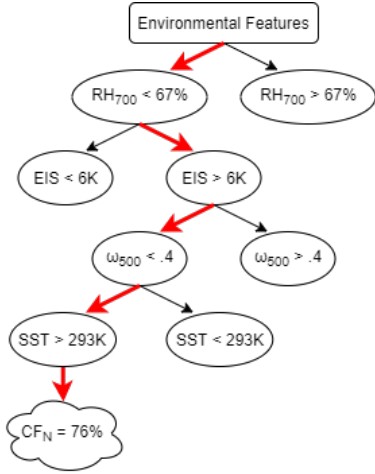

**Figure 1.** An example of how the ML models used derive a prediction from a single decision tree based on the environmental features given as inputs. All ML models used are a collection of these trees, which are created using different pruning, subsampling, and loss (error) functions.

with some amount of subsampling to reduce overfitting. The XG regressor is optimized to penalize incorrect fits of the tree, increasing the overall accuracy of the model without overfitting (Chen et al., 2015).

All (RF, SGB, XG, and MVLR) models are only trained on 80% of the data to avoid overfitting (Train data), leaving out 20% of the clean scenes as a testing sample (Test data). The models are cross validated by re-training on a different subset of 80% of clean observations over ten iterations as a cross validation step to reduce possible sampling bias. The SGB and XG models are further optimized to avoid overfitting by subsampling on a different subset of 80% and 90%, respectively, of the training samples during each iteration of training. Overfitting of the models is diagnosed in section 3.3 of the results by evaluating

mean squared error of the testing subset (20% of the clean observations withheld during training). It is possible that within some specific meteorological regimes, the pristine or polluted aerosol conditions may not be as well represented, and therefore not well sampled within the test/validate/train datasets. Based on prior work ((Douglas and L'Ecuyer, 2019) and (Douglas and L'Ecuyer, 2020)), we believe that for most meteorological regimes the overlap between meteorological states for the pristine and polluted conditions will not be an issue.

## 4    Results

### 4.1    Predicted Changes in Warm Cloud Fraction

Global cloudiness may have increased by $\sim 1.27\%$. All models, from the simple ML to the MVLR, predict a global increase in warm cloud fraction from the simulated pristine, pre-industrial conditions to present day times (Figures 2, 3). The regions with the largest predicted increases, in the southern ocean off the coasts of south America and west Africa, are known to



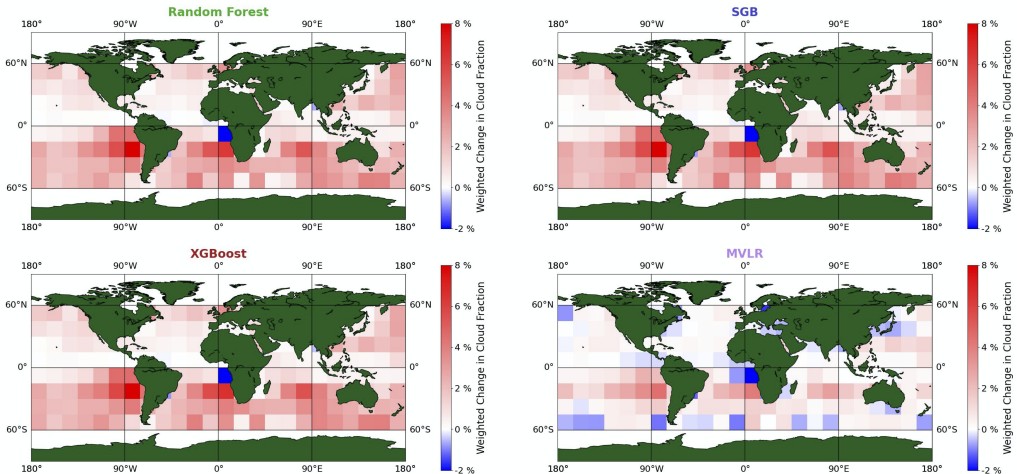

**Figure 2.** The change in cloud fraction (in %) weighted by warm cloud occurrence at a 96 km resolution.

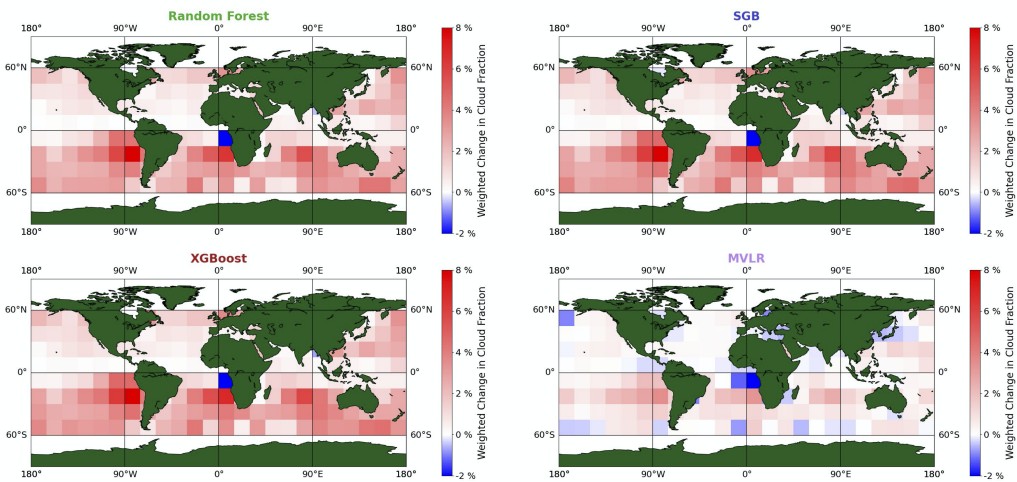

**Figure 3.** The change in cloud fraction (in %) weighted by warm cloud occurrence at a 12 km resolution.

have abundant marine stratocumulus. This abundance and susceptibility, rather than the change in emissions in the southern oceans, drives this large increase. The regional patterns of the ML derived changes agree well between the 12 km and 96 km resolutions. The three ML models, even with different forms of loss functions, subsampling, and pruning of the decision trees, agree well with each other on the regional variation in the signs and magnitudes of the changes. The maximum increase is in the southeast Pacific, a region known to be extremely sensitive to anthropogenic aerosol and therefore widely studied (Wood et al.,

2011). The region with the second largest increase is the south Atlantic, a region known for extensive marine stratocumulus





decks and interactions with anthropogenic emissions from biomass burning (Redemann et al., 2021). The MVLR does not show the cohesive regional patterns apparent in the random forest and SGB estimates and fails to capture all but the strongest signals learned by the ML models, predicting large increases in the southeast Pacific and south Atlantic, with varied responses elsewhere.

The ML model predicted increase in cloud fraction when weighted by occurrence of warm clouds over the observational period varies from 1.17% to 1.37 %, depending on observational scale (Table 1). The ML models agree at 12 km the cloud fraction may have increased by $\sim$ 1.3% and at 96 km the cloud fraction may have increased by $\sim$1.2%, though the scale at which the observations are aggregated has slightly adjusted the magnitude of the predicted increase. The MVLR, in comparison, underestimates the global change at .33% and .34% at 12 km and 96 km scales, respectively. Due to the almost random nature

of most signals predicted by the MVLR, the true increase in global cloudiness from pre-industrial to present day times is likely closer to the ML derived estimates at $\sim$1.27%.

     All four models, from the simple ML to the MVLR, agree that one region in the south Atlantic has likely experienced a decrease in warm cloudiness from pre-industrial to present day times, agreeing with our prior work which analyzed the cloud adjustment response with environmental constraints (Douglas and L'Ecuyer, 2020). This region experiences a multitude of

weather and aerosol phenomenon, from Saharan dust to the subtropical African jet (Adebiyi and Zuidema, 2016; Sauter et al., 2019). The conditions that lead to more expansive warm clouds may be decreasing in occurrence while at the same time conditions in this region are prime to induce cloud desiccation processes such as entrainment-evaporation feedbacks (Small et al., 2009).

     We quantify only the changes for scenes which were already cloudy, as we do not want to possibly conflate cloud feedbacks

(how cloud formation has changed due to global warming) with aerosol-cloud interactions (how aerosol loading has increased or decreased cloud extent.) However, even with this limitation, our models have succeeded in identifying a reduction in warm cloud cover in the south Atlantic, in part because our observational record so far includes this signal. A limitation of ML, and all observationally based studies of climate interactions, is that they can only be used to capture processes and signals seen in the current climate record. It is possible in future climates that more regions will experience the same conditions that have

led to a reduction of cloud fraction seen at this south Atlantic region, however until our we observe these states, they can only be inferred from physically based modeling studies. Vice versa, it is impossible to fully validate studies like Schneider et al. (2019) which approximate a set of future climate conditions and model the cloud response with our set of observations, even with advances like neural networks and other machine learning models.

## 4.2    Forcing due to Changes in Global Cloudiness

It is likely that the increased warm cloud cover has cooled the Earth -0.435 $\pm$ 0.036 Wm$^{-2}$ since the pre-industrial times (Figures 4 and 5) when weighting the estimated change in cloud fraction from the RF, SGB, and XG models equally. The estimated cooling due to this cloud adjustment effect of warm clouds lies well within the range proposed by Gryspeerdt et al. (2016) of -0.1 to -0.64 Wm$^{-2}$ (with proposed estimate of -0.48 Wm$^{-2}$) and within the wider range of the cloud fraction effect proposed by Bellouin et al. (2020) of +0.16 to -1.88 Wm$^{-2}$ (Table 2). Only the simple ML model derived changes in





| Model | ΔCF at 12 km | ΔCF at 96 km |
|-------|--------------|--------------|
| RF | 1.19 - 1.33% | 1.11 - 1.22 % |
| SGB | 1.29 - 1.46% | 1.13 -1.28 % |
| XG | 1.29 - 1.46% | 1.15 - 1.3% |
| MVLR | 0.25 - 0.42 % | 0.26 - 0.41% |

**Table 1.** The predicted change in global, all cloud fraction (in %) with 95% confidence intervals that include the error in prediction in the test set at 12 km and 96 km resolutions for each model.

cloud fraction are used to estimate an approximate change in forcing. Our estimate assumes a constant albedo and ignores any potential increase in brightness due to the Twomey or liquid water path effects. If these were added, our estimate would likely increase, however these effects were not included in our methodology. Unlike the Gryspeerdt et al. (2016) and other studies which define a cloud fraction effect by first defining a sensitivity based on observed aerosol properties, our range has a relatively small contribution of uncertainty due to aerosol, reducing our overall uncertainty in our estimate. The majority of uncertainty

in our estimate lies in our methodology, error from the simple ML models, and instrumentation error of the observations. Our methodology does not hinge on exact measurements of aerosol or exact modeling of changes in aerosol since the pre-industrial times in order to quantify the effect, a relative improvement in the sources of error when quantifying ERFaci. While neural networks, an additional class of ML models that allows for non-linear regression, could have similarly been deployed within, we chose to use stick with decision tree models as they did remarkably well (96% explained variance, below .1% mean squared

error) at predicting warm cloudiness. Further, decision tree models allow you to "peek behind the curtain" and follow a logic as the model calculates a prediction. There may be some amount of uncertainty due to our choice of using decision tree based models, however it is difficult to compare this uncertainty to the uncertainty of using a less transparent type of ML like a neural or convolutional network. The standard error in estimates during the cross validation is included in the uncertainty range to account for the consistency of the model, as some models reproduce results consistently even as the training/testing datasets

change, and others show minute differences in pristine cloud fraction.

The uncertainty range of $\pm.036$ includes error in pristine cloud fraction estimates during testing and variation in the emulated pristine cloud fraction estimates as the model is re-trained in a method called cross-validation. We chose a specific set of cloud controlling factors that have demonstrated skill in predicting pristine cloud fraction. It is possible that using other cloud controlling factors like the estimated cloud-top entrainment index or boundary layer winds may have changed the predicted

pristine cloud fraction, leading to some amount of unquantifiable uncertainty in our analysis due to our choices (Eastman and Wood, 2018; Fuchs et al., 2018). Our estimate does have considerable uncertainty in pre-industrial aerosol estimates. It is possible, by choosing SPRINTARS versus other pre-industrial aerosol models, that our estimated change may be damped in the northern hemisphere. Future work explores how sensitive the change in cloud fraction is to defining a pre vs. present day aerosol limits, or the conditional effect of increasing aerosol using neural networks to infer causal links (Jesson et al., 2021).

We hope in future work to refine the assumptions herein made in order to reduce methodological uncertainty. Additionally not





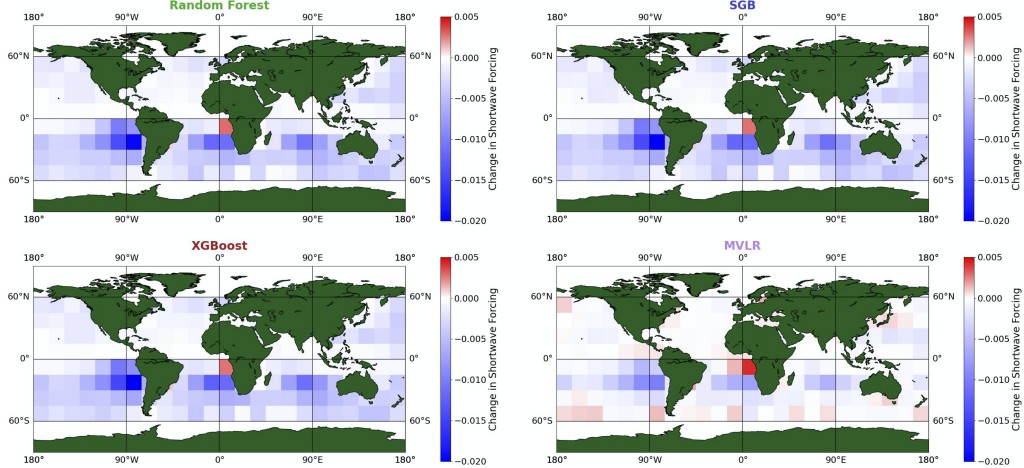

**Figure 4.** The top-of-atmosphere change in shortwave forcing $(\mathrm{Wm}^{-2})$ due to the predicted change in warm cloud fraction for each model aggregated at a 96km resolution and weighted by warm cloud occurrence.

accounted for in our estimate is how aerosol type effects the magnitude or sign of the cloud extent response nor how aerosol may affect the environment. It is possible, through aerosol-environment interactions, that there have been additional changes to warm cloudiness since pre-industrial times. Additionally, aerosol morphology has changed since pre-industrial times, meaning current pristine environments may not reflect the correct types of aerosol present before the industrial revolution. Because

the type of aerosol in present day, pristine environments is not the same as pre-industrial aerosol types, this does lead to some amount of error in our methodology. Further, cloud seeding by anthropogenic activity may have increased the frequency of warm clouds in regions since the pre-industrial times, meaning the weights used in Equation 2 may not reflect the true frequency, or change in frequency, of warm cloud occurrence.

     As with the increases in warm cloudiness seen in Figures 2 and 3, the largest cooling due to increase cloudiness is in the

southern oceans in the southeast Pacific and south Atlantic. The southern hemisphere dominates over the northern hemisphere in terms of cooling and increased cloudiness, likely due to the sensitive nature of these clouds compared to their northern counterparts. Though the northern hemisphere has more emissions due to anthropogenic activities, the lack of expansive, abundant marine stratocumulus like those seen in the southeast Pacific, leads to a smaller cooling effect than in the southern hemisphere. The cooling effect due to cloud adjustments modeled by the simple ML models do not assume a linear, constant

effect, meaning that they can capture the behavior of regions that are fully saturated by aerosol, such as some regions in the northern hemisphere. An advantage of using ML is the pattern recognition of the non-linear interactions in a multidimensional space. While Gryspeerdt et al. (2019) used a piece-wise relationship in order to capture the multiple relationships between liquid water path and aerosol, future estimates of sensitivity could be formed using the non-linear relationships found by ML models, while also accounting for environmental influences on the relationships.





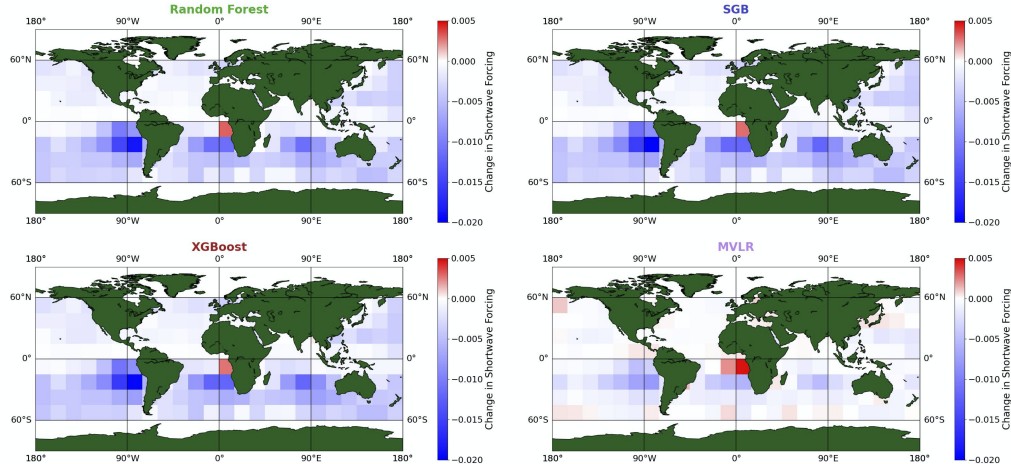

**Figure 5.** The top-of-atmosphere change in shortwave forcing ($Wm^{-2}$) due to the predicted change in warm cloud fraction for each model aggregated at a 12km resolution and weighted by warm cloud occurrence.

| Model | ΔSW at 12 km | ΔSW at 96 km |
|-------|--------------|--------------|
| RF | -0.41 - -0.45 $Wm^{-2}$ | -0.38 - -0.42 $Wm^{-2}$ |
| SGB | -0.33 - -0.50 $Wm^{-2}$ | -0.39 - -0.44 $Wm^{-2}$ |
| XG | -0.44 - -0.50 $Wm^{-2}$ | -0.4 - -0.44 $Wm^{-2}$ |
| MVLR | -0.09 - -0.15 $Wm^{-2}$ | -0.10 - -0.15 $Wm^{-2}$ |

**Table 2.** The predicted changes in shortwave forcing ($Wm^{-2}$) due to the predicted increases in global cloud fraction.

### 4.3 Validation of ML Model Results

The four models can be further validated by investigating their error when tasked with predicting the pristine cloud fraction on a subset of observations held back during their training (Figure 6). The MVLR performs the worst in estimating the cloud fraction in the testing subset. The mean squared error of the three simple ML models all fall below .1% cloud fraction ( .08%, .07% and .08% for the RF, SGB, and XG, respectively) while the MVLR shows over 75x error with an average mean squared error of 6.00%. The simple ML models do exceptionally better than the MVLR at capturing the behavior of clean scene warm cloudiness.

The explained variance scores exemplify how much better the simple ML models perform compared to the traditional MVLR (Figure 7). The MVLR can at best explain ∼50% of the variation in warm cloudiness in clean aerosol scenes, while the simple ML models approach 100% at both 12 km and 96 km resolutions. The explained variance scores decrease as the resolution decreases (from 98.6%, 99.0%, and 98.9% at 96 km to 96.0%, 96.7%, and 94.5% at 12 km for RF, SGB, and XG,





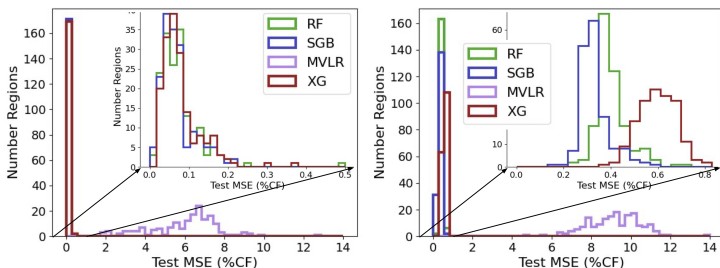

**Figure 6.** The regional mean squared error for each model when tested on unseen observations (20% of the "clean"/"pristine" aerosol scenes) at 96 km (left) and 12 km (right) resolutions.

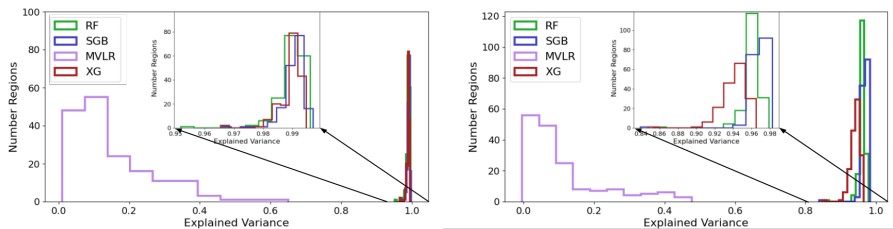

**Figure 7.** The regional explained variance scores at 96 km (left) and 12 km (right) resolutions for each model.

respectively). The explained variance scores not only lend credence to the simple ML models, but to our methodology. Our explained variances also agrees with explained variance scores from Cao et al. (2021) (97%), who used similar environmental inputs in order to derive a non-linear model of warm cloud droplet number concentrations. We assumed that a majority of the variation in clean, "pristine" aerosol scenes are due to the environment, and the amount of explained variance in the cloud 305 fraction explained by just the environment confirms this hypothesis. If our hypothesis had been wrong, and aerosol, even in small quantities, controlled the cloud fraction more than stability, large scale vertical motion, the relative humidity in the free atmosphere, and the sea surface temperature all act to do in conjunction, the explained variance scores would be lower as variation due to aerosol would not be learned by the models. Instead, we see that the regardless of the scale of aggregation of the observations, our simple ML models do exceedingly well at learning the patterns of cloudiness associated with the clean 310 environments (Figures 8, 9).



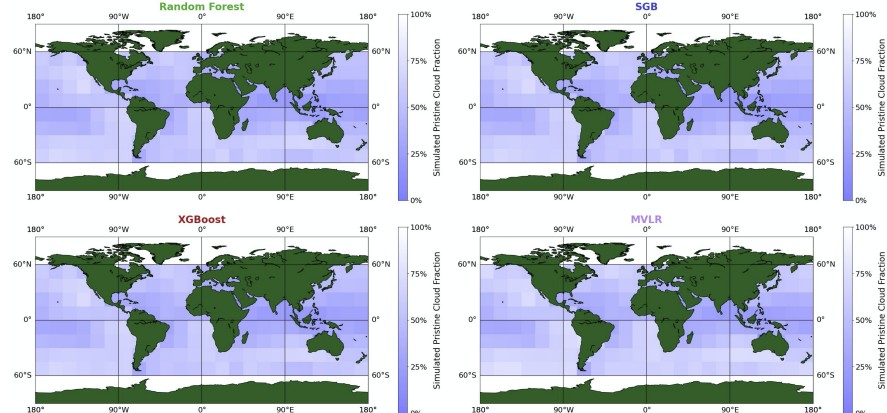

**Figure 8.** The simulated pristine cloud fractions at 96 km observational scale for each model weighted by the occurrence of warm clouds globally.

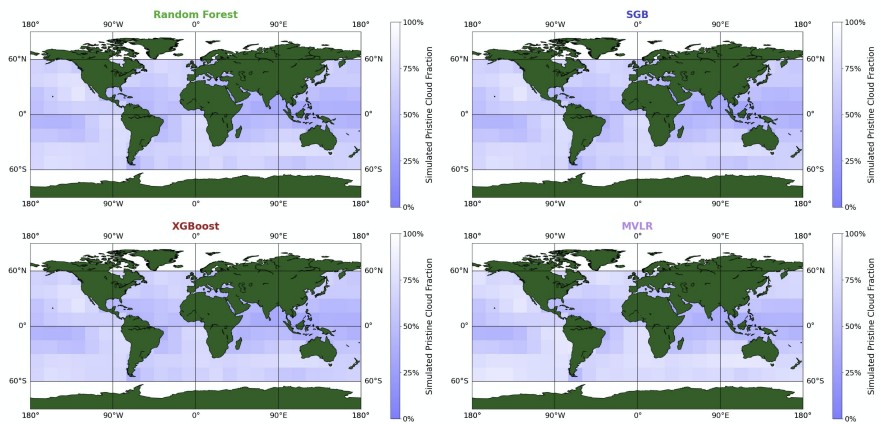

**Figure 9.** The simulated pristine cloud fractions at 12 km observational scale for each model weighted by the occurrence of warm clouds globally.

## 5 Discussion

### 5.1 Beyond linear sensitivities

There are two primary advantages to using ML models to derive the change in cloud fraction: they are not restricted to linear relationships and they do not require explicit proxies for pre-industrial or present day CCN concentrations. Unlike methods which aim to derive sensitivities via linear regression, by finding a difference between the actual observed cloud fraction and observationally based pristine, pre-industrial proxy cloud fraction, signals that are otherwise obscured by non-linear relation-





ships between aerosol and cloud extent. Sensitivity studies, even with environmental constraints, may not capture the correct magnitude or sign of diverse indirect effect processes. The MVLR, which can be considered as emulating approaches that derive a sensitivity to the environment, estimates variable signs in differences compared to the ML models. This is likely
due to the non-linear interactions between different environmental features and cloud extent. While the relationship between aerosol and cloud extent can be non-linear, so too can the relationships between each environmental feature (SST, EIS, etc.) and cloud extent. Some cloud feedbacks may be inherently included in our model estimates as they are controlled by one of our environmental factors (SST, EIS, $\omega_{500}$, $RH_{700}$), however we have not quantified the effects of these nor is our framework the correct way of identifying these feedbacks. Using a ML regression model to derive a pristine cloud fraction circumvents
the necessity of exactly quantifying a true aerosol or CCN proxy to be used to define any type of function between emissions and cloud properties. By aiming to instead define an approximate pre-industrial, pristine aerosol condition cloud fraction, the overall change in cloudiness can be quantified without the additional uncertainty from aerosol retrievals or the appropriate use of clear scene aerosol as a CCN proxy.

The inner mechanics behind the predictions, or the learnings, of the simple ML models can be used to further understand
cloud-environment interactions. The partial dependencies, or how the model cloud fraction estimates depend on certain environmental regimes, reveal complex relationships between different cloud controlling factors and cloud fraction (Figure 10). These relationships can be exploited to not only better understand the physical processes involved in sustaining cloudiness, but as comparative measures for how well our GCMs capture these interactions in pre-industrial runs. How and to what magnitude the environment controls cloud fraction is important to unravel in order to fully constrain aerosol-cloud interactions in GCMs.
If GCMs are failing to accurately capture the unperturbed state of the atmosphere, before anthropogenic aerosol emissions alter cloud properties, then using the differences between no aerosol and aerosol runs to define an ERFaci may be entirely flawed. In our future work, we will explore how the inner mechanics of transparent machine learning methods like decision tree based models can be exploited to identify regimes of cloud controlling factors from observations. We can further use tools like partial dependency plots as shown in Figure 10 to more easily evaluate if ML models are identifying physical, realistic patterns and
how these patterns may relate to similarly derived trends from GCMs.

## 5.2 Judging our Simple ML Models

In the introduction, we proposed two simple, but important, judging criteria for the output of the simple ML models. First, the simulated pristine cloud fractions should be physically realistic, leading to realistic changes in the regional patterns and magnitudes of the warm cloud fractions. Second, the observational scale should not largely impact the change in warm cloud
fraction or resultant forcing, as the model should learn an abstract, scale independent representation of pristine cloudiness. We believe the models presented, the random forest regressor, stochastic gradient boosting regressor, and the extreme gradient boosting regressor, pass these criteria.

The simulated pristine cloud fractions (Figure 8, 9) look physically realistic. The magnitudes of all four models, including the MVLR which has greater test error and a lower explained variance, are plausible. No regions behave like outliers; all
regions show smooth transitional patterns from one to the next, though the models are trained without "knowledge" of where





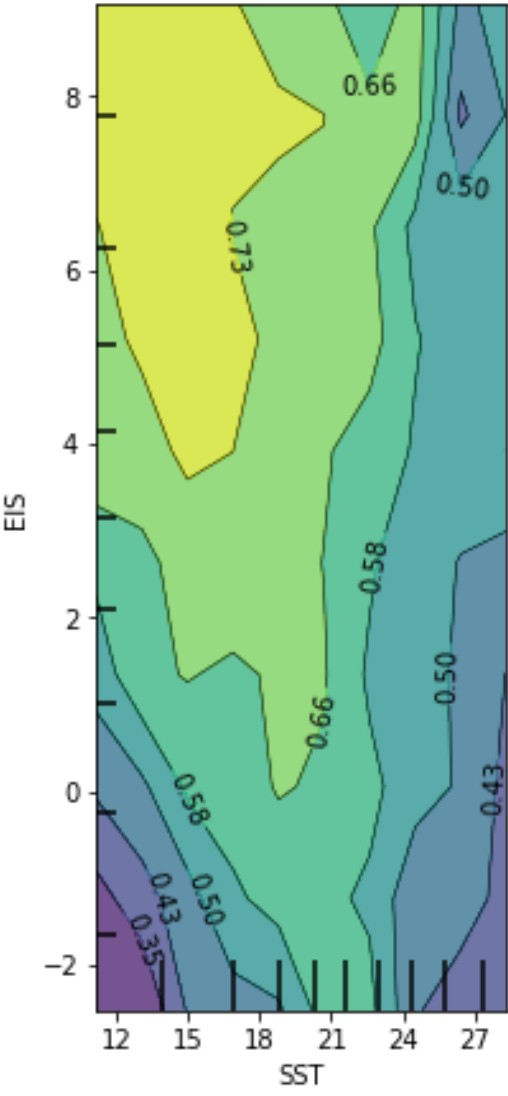

**Figure 10.** An example of partial dependency plots learned by a random forest showing how the predicted pristine cloud fraction (shaded, contoured) varies with the SST (C°) and EIS (K) for all observed ranges of $\omega_{500}$ and $RH_{700}$.

each region lies on Earth. In the future, possible biases due to the reanalysis used could be assessed by determining how using alternate reanalysis (e.g. ERA-5) alters the ML derived responses.

Furthermore, the global mean change in warm cloud fraction is minimally altered by the quadrupling the observational averaging scale (Table 1). The simple ML models converge around a global mean increase of ∼6% when unweighted by
warm cloud occurrence. When weighted by the occurrence of these cloud types in the time period studied, aerosols may have

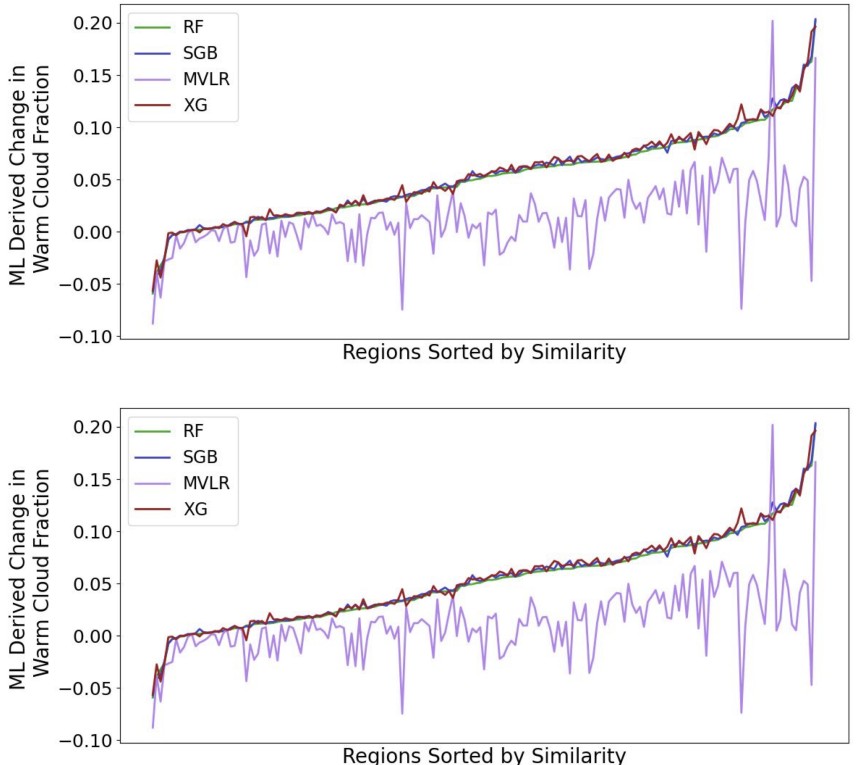

**Figure 11.** The change in cloud fraction sorted by the weighted, mean change in cloud fraction from lowest to highest values regionally at 96 (top) and 12 km (bottom) resolutions sorted by the similarity in the regional change to the overall median change in cloud fraction.

increased the total, global cloud cover by ∼1.27% (1.33%, 1.2% at 12km, 96km). This translates into a cooling effect at the top-of-atmosphere of -.42 (.39 - .46 at 95% confidence intervals) from pre-industrial to present day times. As our results are heavily determined by the pre-industrial aerosol concentrations from SPRINTARS, we believe a set of pre-industrial aerosol concentrations should be agreed upon by those quantifying ERFaci, to reduce the bias these results may have when comparing

against other forcing estimates.

## 5.3 Multivariate Linear Regressions

The difference in the learned signals between the simple ML models and the MVLR is obvious when the regions are sorted by the regional, mean in change in cloud fraction from lowest to highest values (Figure 11). Studies that use an MVLR to understand environment-aerosol-cloud interactions may be capturing at best the linear, unique signals and at worse spurious

signals of these interactions. The stark contrast between the simple ML models and the MVLR along with the sporadic pattern of differences regionally in Figures 2 and 3 are evidence of the latter in some regions.





# 6   Conclusions

Aerosol-cloud interactions may have increased cloud cover by 1.27% since pre-industrial times if the pristine warm cloud fractions predicted by the simple ML methods are an appropriate proxy. The cooling due to the enhanced albedo of the Earth

is estimated to be -0.435 $Wm^{-2}$ ±0.036. This estimate only accounts for the cloud extent portion of the cloud adjustment term of the ERFaci, additional cooling is possible due to LWP and effective radius effects on the albedo. Our results do not include how aerosol may have affected the occurrence of warm clouds globally, which can also alter the warm cloud fraction. Our weighting term does include the occurrence of warm clouds within our dataset, and therefore may be in some way inherently included. In the future, ML could be applied to quantify the heterogeneous effect of aerosol on LWP, cloud optical depth, and

effective radius simultaneously as a multi-targeted prediction in order to unify all cloud adjustments into one model. As seen in Figures 2 and 3, while the global mean cloud fraction response to aerosol can be positive, regional responses, such as those seen off the coast of Africa, are negative as not all warm clouds respond to aerosol loading in the same manner.

The simple ML models adopted here are both physically realistic and scale independent. The patterns of predicted pristine cloud fraction are consistent with the regional environments (Figures 8, 9), while the magnitude of the changes are plausible

given the different cloud regimes in each region (Figures 2, 3). For example, in the trade cumulus region, clouds are less likely to exhibit substantial expansion and increased longevity due to precipitation suppression, and therefore the trades region realize smaller differences in cloud fraction compared to regions with extensive marine stratocumulus such as in the southern oceans. While cloud fraction is a function of averaging scale, the observational averaging scale minimally impacts inferred changes in global cloudiness, nor does it affect the regional patterns. Further, the three models (RF, SGB, and XG) agree well on the

magnitude of the changes for both scales (Table 1) and converge towards a single estimate for the cooling due to changes in cloudiness (Table 2).

Machine learning techniques provides way of performing analysis using satellite observations in order to understand earth system processes (Rolnick et al., 2019). The learnings from ML models can be used to further process understandings and facilitate GCM comparisons. ML appears to have an advantage in reducing the biases due to the observational scale. ML

regression techniques are flexible in that they allow non-linear behavior in a multidimensional space. While we used only four environmental factors in this analysis, more complex analyses could use any number of predictors granted they have some relationship to the predictand. While these models introduce error due to the uncertainty of how they established a prediction, the method itself does not lead to error due to its limitations, unlike those constrained to only find a linear relationship. Knowing a relationship like that between aerosol and many cloud properties is non-linear yet applying a linear regression in order to

derive a sensitivity or forcing estimate leads to unnecessary uncertainty.

*Data availability.*   All data is publicly available through the NASA EOSDIS service. The MVLR, RF, SGB are available using the sci-kit learn package (https://scikit-learn.org/stable/) while the XGBoost is available at https://xgboost.readthedocs.io/en/stable/.





*Author contributions.* AD performed all analysis, data configuration, and wrote the manuscript. TS provided funding, insight, and support.

*Competing interests.* There are no competing interests.

*Acknowledgements.* This project has received funding from the European Union's Horizon 2020 research and innovation program under grant agreement No 821205. Special thanks to the Turing Institute Postdoctoral Enrichment program which provided additional funding.



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
