# Peer review of "Possible evidence of increased global cloudiness due to aerosol-cloud interactions"

_Atmospheric Chemistry and Physics, 2022_

## Referee Comment (RC2)

**Review of:**

**„Possible evidence of increased global cloudiness due to aerosol-cloud interactions"**

**by Alyson Douglas and Tristan L'Ecuyer**

**General comment**

In the manuscript named „Possible evidence of increased global cloudiness due to aerosol-cloud interactions", the authors use machine learning (ML) models trained on a low-aerosol subset of the observational data set to predict warm cloud fraction in these pristine conditions on the basis of four meteorological parameters. The ML models are then applied to predict warm cloud fraction in polluted conditions and used as a pristine aerosol counterfactuals, where the difference between these counterfactuals and the observed warm cloud fraction is quantified as ΔCF and assumed to be due to aerosol-cloud interactions (ACI). The authors then use the ΔCF to estimate an effective radiative forcing due to aerosol-driven changes in warm cloud fraction, which is found to be -.42 W m², in line with other recent estimates.

The topic of constraining ERFaci and the cloud fraction adjustment from observations is highly relevant for climate research (and for the readers of ACP), and as progress constraining ERFaci has been limited over the past decades, new methods of quantifying ERFaci (e.g. exploiting novel ML techniques) should in my opinion be encouraged. While the underlying idea of the authors falls into this category and clearly has merit, I see a number of critical issues and unclarities with the research presented here, and the manuscript leaves the impression of being put together hastily (unit missing, equation missing etc.). Some of the critical issues are

1) sampling issues: The data sets are divided into a clean and a polluted data set, but only trained on the clean data. This is problematic in regions where the aerosol features e.g. a clear seasonal cycle, as data from the polluted season will not be included or underrepresented in the training data. As CF and its meteorological controls have seasonalities as well, realistic predictions of CF during the polluted season can not be expected if this season is largely omitted during training. This is particularly the case, as ML models completely incapable of extrapolation are used. I will go into detail into issues with data sampling below.

2) If I understand the manuscript correctly, ΔCF is quantified for the polluted conditions only (trained on clean, applied to polluted situations?), however, as the clean conditions are also part of today's aerosol distribution omitting them in the ΔCF estimation would lead to an overestimation of the overall ERFaci, as ΔCF would be about 0 in the clean conditions.

3) Lack of transparency/details: The manuscript lacks relevant information into how the models are set up (hyperparameters, sampling strategy of the train/test split). This is important, because this affects the reproducibility of the research, but also because it hides possible reasons for the inexplicably high skill of the ML models in the testing with the independent data.

These aspects are likely to influence the results and conclusions drawn in this manuscript, and as such I cannot recommend this manuscript to be published before major revisions and additional tests are completed.

**Major points**

- Sampling biases

  - Aerosol seasonality: As mentioned in the general comment, I am quite worried about the ML models ability to quantify ΔCF and ERFaci in all regions with a pronounced aerosol seasonality. One example for this would be the Southeast Atlantic, where seasonally overlying aerosol plumes from biomass burning in central Africa are a common feature between July and September/October. During this time of year, CF is highest (stability is highest and SSTs are lowest). If aerosol seasonality is not explicitly considered during the data split, it is likely that training data during the polluted season is sparse, and the highest CF conditions not represented in the training data. It can not be expected that the models perform well in predicting prestine counterfactuals in a season that is underrepresented during training. Considering these aspects, I would expect this issue to be amplified in the Southeast Atlantic, and hence I am not surprised to see this region as an outlier with respect to the estimated ΔCF and ERFaci. I do not agree with the interpretation of the authors given in L.223 – 236 or other text passages in the manuscript interpreting this region.

  - On a similar note, sampling representativeness between clean and polluted data will also be an issue in any coastal region where the aerosol loading is strongly correlated with the outflow of continental aerosols (i.e. dynamics). Low aerosol conditions are unlikely to provide representative environmental conditions to high aerosol conditions in such regions, the ΔCF estimation (and ERFaci) would seem to be less robust here. The authors discuss this a bit in L.195 (f), but discard this issue and it remains unclear to me why they believe this is not a problem. Overall, this study needs to more clearly address and discuss these issues, as they only describe the uncertainties their method reduces but not the ones it introduces.

- Lack of transparency/details, unrealistic model performance: It is really unfortunate that the authors do not provide the necessary methodological details to be able to reproduce and fully understand their workflow and results. Even though the authors consider their machine learning methods to be "simple", their setup is not trivial and its details can influence the results. It is necessary that the authors provide the exact model settings (i.e. hyperparameters), how they are determined, but also provide details on how training and test data is split. This is especially important, as the models perform much better than one would expect for such a complex problem (predicting CF). 98-99% explained variance seems completely unreasonable, and a clear sign of the models overfitting to the data. For example, the explained variance in Chen et al. (2022, nature), who use random forests to predict large-scale CF with **114** meteorologic parameters is 56%. The authors cite a study with comparable skill, however, in that study a highly nonlinear polynomial fit of the fourth power (known to overfit data) is applied to highly aggregated (binned) data, and no independent test skill is reported, so this is just another example of models overfitting the data. Also, the skill increase from the MVLR (0-40% explained variance in the MVLR and > 98% in ML) is suspect to say the least. In my experience, a much smaller performance increase from MVLR to decision trees is to be expected, and this is backed by e.g. Fuchs et

al. (2018, ACP), who report an R² between 0.45 and 0.7 for the decision trees and 0.3-0.5 for an MVLR, and Dadashazar et al. (2021, ACP), who report an R² in predicting cloud droplet number concentration between 0.43 and 0.47 for the decision trees and 0.25-0.28 for an MVLR. One should note that the MVLR results are fairly close to what is reported in this study, however, the ML results are obviously not.

I can speculate that the reason for these high model skills could be that splitting training and test data is done randomly (and not in two completely separated time periods (e.g. 2 years training, 1 year testing)), and that the models then learn the training data by heart. Because training is done on 80% and testing on 20% of the data, and autocorrelation of CF and the predictors is high at the daily time scale, the task becomes very easy for the decision trees, as the test data is not independent from the training data and they are very powerful in learning data by heart. However, this remains speculation, because the authors have not added the necessary information on the model setup in the manuscript.

This (very likely) wrong skill estimate is then used to confirm the hypothesis that "a majority of the variation in clean, "pristine" aerosol scenes are due to the environment" as "the explained variance scores would be lower as variation due to aerosol would not be learned by the models" (L.304 and L.308-309), which in general I would agree with, but not based on the research presented here. It would be much easier to trust the results if the model setup were described completely, and the training and test split done as described above (and considered to be good practice for temporally structured data; Roberts et al. (2017, ecography, doi: 10.1111/ecog.02881)). Additionally, it would be appreciated if the authors would do an additional test on how well the models can predict polluted situations in the year left out for testing.

- It is unfortunate that the authors do not provide any detailed information on the aerosol (MODIS AI) thresholds derived from the SPRINTARS model. The authors only vaguely state that "We choose a selection of scenes with a distribution of AI similar to the pre-industrial values for that region from SPRINTARS [...] (L.103-104)". As the definition of AI values interpreted as pre-industrial proxy is critical for the results, it is necessary that the authors provide a precise statement on how this is determined and ideally a map of this threshold, and information on how much data is in the prestine and polluted groups.

- If I understand the method described in L.150 correctly, ΔCF is only estimated for the "polluted" situations that should represent present-day conditions. However, I don't think that is reasonable, as the „prestine" conditions used as a proxy for pre-industrial conditions are still part of the aerosol distribution in the present-day. Neglecting this leads to an overestimation of ΔCF, because ΔCF should be close to 0 for the „prestine" conditions and thus decrease the overall average ΔCF. Will this make a significant difference in the end? There is no way for the reader to know, as the AI thresholds used are not provided, and hence it remains unclear how much data is assigned to the prestine and polluted groups in each region.

- I see the interpretation of the ML models as „True" and the MVLR as "False" (e.g. L. 182) to be problematic. It is also unfair to compare sorted regional ΔCF values of the 3 ML models to the MVLR to show how similar the ML models are compared to the MVLR (Fig. 11). Clearly the 3 ML models chosen in this study are closely related in the way the learn

(all are ensembles of decision trees), and hence they are expected to more or less lead to the same learned patterns. It would be more interesting to use e.g. a simple neural network as an additional comparison, as this represents a different way of mapping the model input to the output than the decision trees. Also, a simple neural network does not have similar overfitting issues, as it cannot learn training data by heart as easily as the decision trees.

- The authors hypothesize that a "good" ML model will be 1) physically realistic, and 2) observational scale independent. In my opinion, the authors do not show convincing results to support either hypothesis. 1): The only results that can be interpreted as a sign of the models being physically realistic is presented in Fig. 10. However, in the text this is treated as an outlook, and it is not even clear in which region the results were produced ("example"). It is certainly not physically realistic that 4 large-scale environmental controls explain more than 90% of daily CF variability. 2): The authors use two different initial cloud fraction scales (12 and 96 km) which are then both aggregated to 15°x15°. The analysis is then conducted on the same scale, and all predictor data seems to be identical as well. While this is an interesting experiment, it does not seem to be a convincing argument that the results are scale independent.

- While this manuscript is focused on the cloud fraction adjustment to aerosols, this is not reflected by the introduction, where the cloud fraction adjustement and current approaches to quantify it and its radiative forcing are not discussed. Actually, the LWP adjustment is discussed in much more detail. I recommend that the authors provide a better overview of past observational research done on the cloud fraction adjustment and provide sources for the statement in L. 25.

**Minor points**
- L.8: units missing for the cooling estimate
- L.28-31: Cloud fraction retrieval is also not straight forward, as it depends on an optical depth threshold and is affected by surface reflectance etc.. A change in optical depth due to the Twomey effect could thus also lead to an artificial increase in CF for thin clouds (Mieslinger et al. 2022, ACP).
- L157: Equation is missing
- L.171-174: Sentence is quite hard to read.
- L.176-178: Confusing sentence
- L.229-231: I don't quite understand why only cloudy scenes are used, and don't understand the authors reasoning given here: "we do not want to possibly conflate cloud feedbacks (how cloud formation has changed due to global warming) with aerosol-cloud interactions (how aerosol loading has increased or decreased cloud extent.)[sic]", as cloud feedbacks are not constrained to changes in cloud formation, but rather describe the change in overall cloudiness (could also be due to a longer lifetime or larger spatial extent of the clouds).
- L.231-232: Is there a source for the statement that the observational record includes a reduction in warm cloud cover in the south Atlantic? If not, I don't understand how the authors can claim that "our models have *succeeded in identifying* a reduction in warm cloud cover in the south Atlantic". The authors then go on with this claim a few lines further

down: "It is possible in future climates that more regions will experience the same conditions that have led to a reduction of cloud fraction seen at this south Atlantic region"

- L.233-234: Technically, this is not true for all machine learning models, as some do have the capability for a limited extrapolation (e.g. MVLR, simple neural networks), but this is true for tree-based ML models which can only interpolate on the training data.

- L.319: I don't undestand what the authors mean with this sentence.

- L.348-351: The authors use the average model predicted cloud fraction of the prestine situations used for training (Fig. 8 and 9) and interpret this as a sign that the models are consistent with each other and represent physics internally (in the conclusion). This is not related to physics or a sign of consistency of what the models represent at all though, but just the fact that any statistical or ML model will quickly learn the average predictand value during training. I believe that these Figures and the discussion on it (also in L. 308-310 and the conclusions) add nothing to this paper and should be removed.

- Figures 2-5: I would recommend that the range on the colorbars should be equally far positive/negative, as one naturally compares the hue of the colors, but in these figures the change of the hue for positive/negative values is different by a factor of 4.

- The difference between $\Delta CF$ of the Northern and Southern Hemispheres is interesting, and I think this should be discussed with more detail. I would especially be interested if there is a hemispheric difference between the AI threshold used in this study.

**Specific points**

- L.37: „environment" is used as a term for meteorological factors (excluding aerosol), however this is not clear as aerosols are also a component of the environment
- L.52: to constrain
- L.54: Adjust citation style
- L.75: 2 Methods 3 Data
- L.134: Grammar
- L.146: „split into by 20%"
- L.235: „until our we"
- L.321: EIS has not been introduced so far

---

## Author Comment (AC1)

**Main Points**

**Clean conditions as a proxy for pre-industrial**

We agree that a set of "clean" observations defined by a limit in the aerosol index may not be the perfect proxy for pre-industrial conditions, however we view our work as a frameowrk for how this could be done. In the future, combining more limits on other environmental parameters in order to define the "clean", pre-industrial proxy set could help set the clean conditions to better match the pre-industrial environment. If we limit the number of observations in our clean set, we do run the risk of having an unrepresentative "pre-industrial" set of observations as compared to the present day set. In causal inference, this would lead to a positivity assumption, whereby an unperturbed cloud does not have a perturbed proxy or vice versa. To avoid violating the positivity assumption in our work, we chose to only set one constraint to define our "clean", pre-industrial proxy, as other methods would require more abstract machine learning methods such as represenation learning that are less interpretable and more prone to error due to the model chosen (Tec et al. 2022).

We have added as a caveat to our methods section: "It is likely that our clean set does not perfectly reflect the true nature of pre-industrial clouds. Creating counterfactuals requires us to not violate the positivity assumption, whereby the clean set of observations the models are trained on are representative of the possible conditions contained in the polluted scenes. In the future, representation learning similar to Tec et al. 2022 could be used to create a better representation of the pre-industrial environment without violating this base assumption."

**Southern hemispheric change**

This is likely due to two factors:

1. Warm clouds are more likely in the southern hemispher and therefore have larger changes when weighted by their occurrence.
2. The regime of stability and sea surface temperatures of the southern atmosphere cloud decks leads to larger sensitivities.

As shown in Figure 10 (right), there is a large regime of cool sea surface temperatures and high stabilities that exhibits the largest cloud fractions. This regime most likely coincides with the clouds of the California stratocumulus, the region of most frequent warm clouds in the northern hemisphere. These clouds are likely less sensitive as their cloud fractions are already extremely high and in that sense they are at a "saturated" point of cloudiness. Though their difference in sea surface temperature is only a few degrees (see map and trends of two regions below), this few degree difference shifts the California region to a more cloudy regime that is less sensitive to aerosol even at the same stability.

[Figure]

[Figure]

Uncertainty Range

We agree that it is optimal to be transparent about our uncertainty range, as it is not the same as an uncertainty range from an Earth system model or sensitivity analysis that then compounds uncertainty. We have repeated our uncertainty limitations in the forcing section of the results:

"Not measured within this uncertainty range is how our technique, and the imperfect use of AI limits as a proxy for pre-industrial esque versus present day clouds, leads to either an over or underestimation of the effect. This type of epistemistic knowledge is difficult to quantify; in the future, applying "uncertainty aware" models that quantify how unknown factors may alter the prediction could remedy this fault (Jesson et al. 2020)."

**Minor points**

L12: Updated to include newest IPCC report section on Short-lived Climate Forcers.

L28: Removed Rosenfeld and replaced with opportunistic/natural labs reference as that demonstrates the spread in LWP estimates better than two conflicting papers.

L28-31: Rephrased to "Similarly, although increased cloud extent should in theory be an obvious to observe, due to confounding relationships it remains difficult to isolate the change in cloudiness due anthropogenic aerosol from environmental changes."

L40: Added: "Andersen et al. 2017 set some groundwork in using a simple artificial neural network (ANN) to understand cloud properties, but was limited by the shallow depth of the network (2 layer ANN) in fully capturing and translating the global drivers... Our ML models differ from an ANN as decision tree models can have variable depths dependent on the data itself, allowing for better representation of multivariate, non-linear interactions like those between clouds and the environment."

L70: Rephrased to "a majority of the uncertainty" as our methodology itself is the main source of uncertainty, not the use of AI/AOD/Nd as a proxy for anthropogenic aerosol.

L75: Fixed to subsection.

L80: Fixed to only have AI defined as an acronym once.

L143: A higher resolution version of SST from AMSR-E is available as a CloudSat auxillary product. Added: "...available from the AMSR2-AUX CloudSat product." to clarify the source of the data.

L156: Yes, we have clarified this definition of SW CRE.

L229: Reworded to: "We quantify only the changes for scenes which were already cloudy, as we believe delineating how cloud fraction may have increased is different in nature than how cloud occurrence may have changed due to anthropogenic aerosols."

L267: This was included due to feedback that SPRINTARS results may differ from other models' results.

Figure 4 Caption: These results have been weighted by the occurrence of warm clouds within each 15° x 15° region.

L287: Yes, Gryspeerdt et al. 2019 used a joint distribution, but ultimately found a sensitivity by fitting two linear relationships depdent upon "low" and "high" values of Nd. (https://acp.copernicus.org/articles/19/5331/2019/#section2)

L305-308: Reworded to: "If our hypothesis had been wrong, and aerosol's significance to cloudiness was greater than the environment's, the explained variance scores would be lower as variation due to aerosol would not be learned by the models. Instead, we see that our simple ML models do exceedingly well at learning the patterns of cloudiness associated with the clean environments (Figures 6, 7)."

L322: Rephrased to: "Changes in cloud occurrence, i.e. whether aerosol or climate change lead to more or less clouds developing, are difficult to quantify and remove from our estimates. In the future, finding a way to evaluate how cloud occurrence itself may have change would be invaluable to understanding ACI."

Fig 11: This was an error (96 vs 12 written in Latex) and has been corrected. The two images are now correct, though they do look alike.

**References**

Tec, M., Scott, J., & Zigler, C. (2022). Weather2vec: Representation Learning for Causal Inference with Non-Local Confounding in Air Pollution and Climate Studies. arXiv preprint arXiv:2209.12316.

---

## Author Comment (AC2)

**General Comments**

The authors trained the models with 4 parameters: boundary layer stability, relative humidity, vertical motion, and sea surface temperature. Why did they use these 4 parameters? Are they relevant for all the different regions? I think that there are too few parameters to determine the cloud fraction.

To help clarify to the reader before the methods section, our particular set of environmental predictors is introduced, we have added to the introduction:

"In particular, the stability of the atmosphere, often indicated by the estimated inversion strength or lower tropospheric stability, and the humidity of the free atmosphere that the boundary layer clouds entrain strongly control the sensitivity of the cloud fraction to aerosol perturbations (Gryspeerdt et al. 2016). Further, others have indicated the presence non-linear, sudden transitions of warm clouds due to their dependence on a strong inversion and cool surface temperatures to maintain equilibrium (Schneider et al. 2019)."

These parameters have been shown to be highly correlated with cloud fraction in past studies, such as EIS explaining 70% of the variance in major stratocumulus regions globally (Wood & Bretherton, 2006). As mentioned within the Methods section, all four parameters are known to be cloud controlling factor (Klein et al. 2017). Given the models show high explained variances and low mean squared errors on the test data, it is unnecessary to add additional parameters to our dataset. Further, the more environmental variables the models are trained on, the higher the likelihood of violating the positivity assumption, whereby for every possible environmental regime, there must exist a corresponding counterfactual representation of that regime. By limiting our parameters to the strongest cloud controlling factors, we are reducing the likelihood of violating this assumption.

Do they consider all clear sky pixels around the clouds to do the interpolation, or the closest clear-sky pixels to the considered cloudy pixels? If the clouds are horizontally extended, is there a maximum distance between the cloudy pixel and the clear-sky pixels?

We remove the AI nearest the clouds (within 2 km) as mentioned within the Methods section. This reduces the chances of hygroscopic growth from influencing measurements of AOD or the Angstrom exponent. Assuming the small likelihood that clouds are affecting AOD measurements beyond 2 km however would not substantially change our results as we do not use AI as a quantitative metric beyond delineating "clean" from "polluted" environments, and those affected scenes would then more likely be delineated as "polluted," perhaps incorrectly. The predicted cloud fraction for these scenes would most likely match the observed cloud fraction, as our cloud fraction estimates are based on active retrevials from CloudSat and will not be biased by near cloud hygroscopic growth of aerosol. Hygroscopic effects are most likely to alter results when AI is used as a direct, quantitative proxy for CCN to define sensitivities, as this can bias the high AI conditions.

All the considered results are shown averaging over time and space, are the models still good with a single prediction? There is no measure to determine if the results are statistically robust or not even on the figures. Are all pixels show a statistically significant change in CF?

Decision tree methods are meant to be used for tabular data, such as our satellite snap shots. Our data is not cross-sectional, as that would imply we use only a single time for each point. Instead, inherently within our data, best captured by the environment, are the fluctuations over time. As the satellites pass over each point, they add a new shap shot of warm clouds for us to analyze. By including four years of satellite datea (2007 - 2010), we can then more robustly understand these variations as a function of the environments they occur in, environments which change over time.

All of the simple ML models showed statistically significant differences, however that is not the type of metric that should be used to validate a study like this. If the predictions were incorrect, then the results could still be statistically significant as determined by a single p-value. We chose to show the explained variance scores and mean squared errors to justify that our models were correctly capturing the behavior of clean clouds and that the error in those predictions is smaller than the predicted changes in cloud fraction.

The authors performed the training of the models for different regions. Some regions are constantly under the influence of anthropogenic aerosols or maybe not after specific events (e.g., precipitation) but is it safe to consider this as pre-industrial cases? Also, the regions are never properly defined and I do not know what they refer to.

We have added additonal clarification of how our assumptions may leade to uncertainty in the methods and results section.

The effects of aerosol on precipitation occurrence, duration, or severity remain uncertainty; to remove any uncertainty of aerosol-cloud-precipitation interactions we only consider non-precipitating scenes within our analysis. We consider our predicted, clean cloud fractions as a proxy for pre-industrial, however we acknowledge several times within our manuscript that this has limitations. We do not consider what we are predicting as the true pre-industrial cloud fraction for each region, as the global environment, including large scale dynamics determining local meteorologies, has changed due to global warming. However, our proxies are appropriate to create a base estimate of how cloudy the Earth may have been. As stated within the discussion, we believe our technique is a more accurate method than what is currently being done, i.e. using the product of linear sensitivities and the assumed, regional change in AI (or other CCN proxy) to find the overall change due to aerosol-cloud interactions.

The regions are defined in the Methods section under Data and under Methodology. Additional references to what the regions refer to (the 15° x 15° aggregation grid of observations) have been added throughout the manuscript to clarify their resolution.

The authors state that they assumed a constant cloud albedo and therefore cannot account for the Twomey effect. Therefore I am wondering why estimating the radiative forcing at all?

This is correct that we do not calculate a Twomey effect, as we do not compare the albedo of the polluted vs. clean or predicted clean to find how the albedo has increased due to aerosol interactions. The Twomey effect is only one portion of aerosol-cloud interactions that leads to an increase in forcing. An increase in cloud amount will also leading a change in forcing, as more cloud will then reflect more incoming sunlight, cooling the Earth.

We are able to accurately calculate this forcing by using the CERES fluxes. The radiative forcing at the top-of-atmoshpere from CERES inherently includes cloud albedo as brighter scenes -> a higher SW TOA.

```
SW TOA = SW CRE + SW CLR
whereby when fully expanded
SW CRE = Incoming Sunlight x Cloud Albedo x Cloud Fraction
```

Using the SW CRE from CERES includes any albedo changes due to aerosol.

In the figures, I do not understand why the resolution of 12km and 96 km have not been plotted instead of the 30 degree x 30 degree boxes. Also the changes in cloud fraction are shown but I am wondering if the changes are statistically significant for every box (see point 5).

The changes are statistically significant for every region (p << .05). CloudSat is an active sensor with a native resolution of ~ 1 km x 1 km. In order to define a "cloud fraction," you must choose a resolution to average cloud occurrence over. Therefore, to understand how this choice of resolution (aka aggregation of observations over a chosen scale) influences either the training/predictions of the ML models or the estimated change in cloud fraction, we compare our observations averaged over 12 km along track and 96 km along track. These along track observations are then re-aggregated over 15° x 15° regions in order to train regional models and create regional predictions of the "clean" cloud fractions (at both 12 km and 96 km resolutions).

I think the ML techniques are useful for a deeper analysis as presented in Figure 10 to understand the correlation between the different parameters. Otherwise I feel that this paper is presenting a promising method (then ACP might not be the best journal) with interesting results on CF but I really would like to see an explanation on the reasons for the CF changes, if one parameter is more important than others, if it is the same for all regions...

We agree that understanding the different cloud controlling factors and their interactions with each other is important and have already written up our additional, related work to JGR: Atmospheres (*in review*). However, this work we believe does fit within the bounds of ACP as other similar articles (such as Fuchs et al. 2018) and new methodologies (Gryspeerdt et al. 2021) that deal with understanding clouds, aerosols, and the envrionment have been published before.

I am not sure the study mentions exactly what a region is, is it a 30x30 degrees region ?

This is detailed in the Methodolgy. We have added additional reminders of the gridding resolution to the results section to help clarify our regional resolution.

Machine learning models require some parameters as input, for example the depth of the tree, boosting iterations, learning rate for random forest regressor. The current study does not mention the parameters used and how they are decided. Some details are required.

We have added the details of the hyperparameter tuning for each model. As these are all decision tree based models, we tried to keep the depth consistent. Some models (SGB, XGBoost) allowed for sub-sampling, which we utilized in order to help evaluate if the Random Forest was overfitting (as the SGB or XGB would then have shown different results). We have added to the Methods section: "The RF hyperparameters include having bootstrapped training, meaning the RF is trained on a random subset of observations over each iteration, the number of iterations was limited to 125, and the depth limited to 30. The SGB hyperparameters are a max depth of 30, a learning rate of 0.1, a minimum number of samples per

leaf as 3, subsampling at 0.8, and the number of iterations as 200. The XG hyperparameters are max depth of 30, subsampling of 0.8, 'GBTree' as the booster type, a histogram tree method, a learning rate of 0.8, and the max number of iterations to train as 300. These hyperparameters were chosen by using a grid search over each model using as subset of the total, global data. "

**Minor comments**

**Specifics:**

Line 1: Uncertainty is also gramatically correct.

Line 15: Aerosol can be used in place of aerosols in this context.

Line 15: Sentence now reads "Aerosol enters a cloud and acts as cloud condensation nuclei (CCN), increasing the total number of cloud droplets."

Line 25: Aerosol can be used as is in this context.

Line 54: Fixed, removed paranthesis.

Line 106: "in" added to sentence.

Line 122: Changed to affect.

Line 145: Removed "a."

Line 254: Fixed, changed wording to "proceed with."

Added new IPCC report reference, specifically as ACI are now contained within a "short lived forcers" section.

Fixed section headings between Introduction and Methods.

Added additional citations for data used, specifically the MODIS Deep Blue Aerosol product and AMSR-E Sea Surface Temperature validation.

Yes, we omit multilayer scenes and focus only on single layer, marine warm clouds. We have added a more clarifications of this to Data section.

line 156: We have removed warm from the equation and added multiple clarifications that our cloud fractions are the single level warm cloud fractions within each region.

Equation 2: Fixed.

line 169: We agree and have rephrased that statement to read: "To a first degree, warm clouds dominate the cooling due to ACI globally; however polar low clouds could potentially also have some small cooling effect (Christensen et al. 2016, Rosenfeld et al. 2014)."

L.190 You are correct, this was an error. We have added when specifying the hyperparameters of each model: "Subsampling over each iteration, as the XG and SGB allow, reduces the chances of overfitting as this forces the trees to generalize; the RF uses random sampling over each iteration for similar purposes."

The part "Validation of ML Model Results" arrives at the end of the paper. Should it not be more fitted at the beginning of the result section? Or it should be stated that a discussion on the validation is later when presenting the method.

We have rephrased the section to "Validation of ML Models against Our Criteria" as we set out two specific criteria in the introduction that we would use to judge our results (resolution independence and agreement between models). The focus of the study is two folds, therefore the results have two focuses: the science focus of how aerosol may have increased cloudiness and the ML focus on how to create a set of accurate models agree with each other.

Figure 6: That is .4% not 40%. The zoomed in portion shows values below 1%, hence all percentages are a fractional percentage.

From lines 301 to 310: We have rephrased the paragraph. The core of the paragraph now simply reads:

"The explained variance scores not only lend credence to the simple ML models, but to our methodology, as a core assumption of our methodology was that the environment in "clean" scenes could explain a majority of the variations in cloudiness. "

Figure 11: We have updated the caption to read: "The change in cloud fraction (y-axis) for each region sorted by their similarity (x-axis) at 96 (top) and 12 km (bottom) resolutions . The change in cloud fraction is weighted by the regional occurrence of warm clouds. The similarity is found by sorting the median change in cloud fraction from the ML models from smallest to largest values. This allows us a unique viewpoint on how the ML models compare to the MVLR."

**References**

Klein, Stephen A., et al. "Low-cloud feedbacks from cloud-controlling factors: A review." Shallow clouds, water vapor, circulation, and climate sensitivity (2017): 135-157.

Wood, Robert, and Christopher S. Bretherton. "On the relationship between stratiform low cloud cover and lower-tropospheric stability." Journal of climate 19.24 (2006): 6425-6432.

---

## Author Comment (AC3)

**Major Points**

**Hyperparameters**

Based on comments from this review and other reviewers, we have added details on the setup of each of the models including whether the model used subsets to help with overfitting.

"All (RF, SGB, XG, and MVLR) models are only trained on 80% of the data (Train data), leaving out 20% of the clean scenes as a testing sample (test data). The RF hyperparameters include having bootstrapped training, meaning the RF is trained on a random subset of observations over each iteration, the number of iterations was limited to 125, and the depth limited to 30. The SGB hyperparameters are a max depth of 30, a learning rate of 0.1, a minimum number of samples per leaf as 3, subsampling at 0.8, and the number of iterations as 200. The XG hyperparameters are max depth of 30, subsampling of 0.8, 'GBTree' as the booster type, a histogram tree method, a learning rate of 0.8, and the max number of iterations to train as 300. These hyperparameters were chosen by using a grid search over each model using as subset of the total, global data. Subsampling over each iteration, as the XG and SGB allow, reduces the chances of overfitting as this forces the trees to generalize; the RF uses random sampling over each iteration for similar purposes. The models are cross validated by re-training on a different subset of 80% of clean observations over ten iterations as a cross validation step to reduce possible sampling bias."

If the models were overfit and simply memorized the training data as you suggest, this should have greatly decreased their performance during cross validation. Further, the models do not experience a decrease in their mean squared error or explained variance scores in the test set relative to the training set until the test set is >40% of observations. The other ML based studies predicting cloud fraction cited (Fuchs et al. and Chen et al.) use vastly different cloud fraction estimates. Our observations, as explained in the data section of the methods, come from an active satellite radar, which is much better at separating low, warm clouds from other cloud layers and only reports its data in along-satellite track segments of ~1 km x 1 km. MODIS observations are from a passive sensor, which changes the nature of the observations, their accuracy in defining warm clouds vs. other cloud types, and the accuracy of defining cloud vs. aerosol scenes. It is likely these differences lead to the difference in skills. In Cumulo: A Dataset for Learning Cloud Classes, we achieved 89% explained variance scores using only a single year of CloudSat/CALIPSO data when predicting a much more convolved, intricate problem. Higher resolution observations lead to a more precise, accurate estimation of the environmental drivers of cloudiness.

**SPRINTARS to define pre-industrial**

A previous version of this same study set the pre-industrial AI to 0.08 as the limit. The reviewers on that version of this paper did not agree with setting a single AI as the pre-industrial threshold. With that feedback in mind, in this version of the paper we implemented a moving

threshold based on the regional pre-industrial aerosol index from the SPRINTARS model. This is the same pre-industrial AI used in previous studies to define the forcing from ACI (Douglas & L'Ecuyer 2019, 2020) and others looking direct effects (Matus et al. 2019). A more map of the AI used can be found in Douglas & L'Ecuyer 2019. This threshold is much lower than the previous limit of 0.08 and led to a slight increase in the change in cloud fraction estimates.

[Figure]

The regions with a higher limit are those off of the coasts of desert regions, such as near saharan

[Figure]

Africa and western United States.

The number of observations coincides with where low clouds lie. This is also what leads to the greater weighting of the southern hemisphere as all forcings are weighted by occurrence.

**Change in CF**

The change in cloud fraction is found for all scenes, however since the predictors are trained on the pristine scenes, those predictions would therefore "cancel" each other out as the predicted pristine and actual observed pristine would be the same. The change in cloud fraction is then not overly weighted towards the polluted observations.

**MVLR**

We did implement a simple (two layer ANN) and complicated (4 layer, dense) neural networks as comparisons when beginning this work. The neural networks performed worse than the decision trees. The drawbacks of adding layers and steps which decreases our interpretability of the neural network were outweighed by their decrease in accuracy compared to the more simple decision tree architectures chosen. Neural networks excel at images, segmentation, and classification. Decision tree models are still the state-of-the-art standard however for tabular data. It is possible that for some of the passive satellites such as MODIS or GOES, which

create data "images" rather than segments like CloudSat, a neural network would perform better given the type of data.

**Minor Points**

L8: Units of Wm-2 added.

L28 - 31: We have added: "And though cloud fraction is a physical state that can be directly observed by our naked eye, our sensors aboard satellites must still assume thresholds on values like optical depth to determine if a scene is truly"cloudy" (Mieslinger et al. 2022)."

L157: Fixed.

L171 - 174: Reworded to: "Likely the pristine scene conditions are minimally, if at all, affected by aerosol direct effects, therefore the only uncertainty or bias direct effects may pose are in extremely polluted scenes which experience a decrease in cloud fraction due to aerosol absorption and atmospheric heating."

L176 - 178: Fixed the strange wording: "A random forest regressor (RF), stochastic gradient boosted regressor (SGB), and an extreme gradient boosted regressor (XG) are compared against results from a multivariate linear regression."

L229-231: With feedback from this review and others, we have reworded our explanation on why we only predict for clouds which have formed. In essence, we treated aerosol effects on cloudiness as two distinct problems: how aerosol may alter the amount of cloud (which we can quantify) and how aerosol may alter *when* cloud forms (which is harder to prove). Further, altering when cloud forms can be influenced by large scale feedbacks. This now reads: "We quantify only the changes for scenes which were already cloudy, as we believe delineating how cloud fraction may have increased is different in nature than how cloud occurrence may have changed due to anthropogenic aerosols."

L231-232: We have added a reference to a similar study using the same data from CloudSat which found a reduced cloud albedo in the south Atlantic. A recent study by Zhang and Feingold 2022 (*in discussion*, https://acp.copernicus.org/articles/23/1073/2023/) has also found distinct decreases in albedo and extent in the south Atlantic. Our past work (Douglas & L'Ecuyer, 2020), using a different framework to separate the effects of aerosol on cloud extent vs. brightness, also found a decrease in extent in the south Atlantic.

L319: Simplified to: "The MVLR, similar to historical methods of estimating the sensitivity of a cloud property to a CCN proxy, displays variability in the sign and magnitude of the change in cloud fraction compared to the ML models."

Southern Hemisphere: We have added more discussion on the magnitude including how our weighting leads to these results in the discussion.

Specific points:

"environment" We have clarified: "In our context, we term the environment to mean the local meteorology of the cloud without considering aerosol as an environmental feature."

"to constraint" fixed to "to constrain."

2 Methods 3 Data fixed

"split into by 20%" removed by

"until our we" removed our

L321: fixed to read stability

**References**

Douglas, A., & L'Ecuyer, T. (2019). Quantifying variations in shortwave aerosol–cloud–radiation interactions using local meteorology and cloud state constraints. Atmospheric Chemistry and Physics, 19(9), 6251-6268.

Douglas, Alyson, and Tristan L'Ecuyer. "Quantifying cloud adjustments and the radiative forcing due to aerosol–cloud interactions in satellite observations of warm marine clouds." Atmospheric Chemistry and Physics 20.10 (2020): 6225-6241.

Matus, Alexander V., Tristan S. L'Ecuyer, and David S. Henderson. "New estimates of aerosol direct radiative effects and forcing from A-train satellite observations." Geophysical Research Letters 46.14 (2019): 8338-8346.